# K-PGD: Fast Discrete Projected Gradient Descent with K-Means Acceleration on GPT

## Abstract

Projected Gradient Descent (PGD) is a workhorse for optimization over discrete sets, but with large vocabularies the projection step becomes the runtime bottleneck. We present *K-PGD*, a $k$-means–accelerated variant that replaces exhaustive projection with a centroid-based shortlist followed by a restricted search. The approach provides simple per-iteration certificates that quantify approximation error and yield convergence guarantees for PGD with approximate projections. Our theory connects cluster geometry to certificate strength and gives iteration bounds under bounded accumulated error. In a GPT-2 token-substitution case study, K-PGD reduces projection cost while preserving attack success and solution quality, showing that clustering can substantially accelerate discrete PGD without compromising rigor.

## 1 Introduction

Projected Gradient Descent (PGD) is a cornerstone algorithm for constrained optimization, widely applied in adversarial training (Madry et al., 2019), robust optimization (Ghadimi et al., 2016), and machine learning more broadly. Its effectiveness arises from alternating between gradient descent and a projection step that enforces feasibility. While the convergence theory of PGD is well developed in convex (ber, 1997; Beck & Teboulle, 2009), nonconvex, and stochastic regimes (Schmidt et al., 2011; Davis & Drusvyatskiy, 2018), the projection step remains a computational bottleneck—especially in high-dimensional and discrete domains. This challenge is particularly acute in natural language processing. In adversarial NLP attacks (Alzantot et al., 2018; Ren et al., 2019; Zhao et al., 2018), PGD must project updates onto a discrete vocabulary embedding set with tens of thousands of tokens. Unlike continuous $\ell_p$ constraints, this projection reduces to a high-dimensional nearest-neighbor search, often more costly than gradient computation itself. For large language models (LLMs), exact projections can be prohibitively expensive, limiting the practicality of PGD.

We address this issue by introducing a $k$-means envelope acceleration scheme for PGD with discrete projections. The feasible set (e.g., vocabulary embeddings) is partitioned into clusters, each represented by a centroid and radius. These clusters serve as envelopes that bound inner products between the gradient direction and cluster members, allowing efficient screening: only clusters with competitive bounds are searched in detail. This shortlisting mechanism yields substantial runtime reductions. Our framework provides theoretical guarantees. We introduce $\delta$-*proximal certificates*, which quantify the error of approximate projections induced by clustering. These certificates align with the inexact PGD framework (Schmidt et al., 2011), ensuring that accelerated PGD retains descent guarantees. Specifically, we show that the score gap between retained and discarded clusters defines a computable per-step error bound, yielding global convergence with an additive floor tied to clustering quality. This analysis provides, to our knowledge, the first link between $k$-means clustering theory (Kannan et al., 2004; Awasthi & Sheffet, 2012) and projected gradient methods.

We validate our method on adversarial attacks against GPT-style language models, where projection onto large vocabularies is the primary bottleneck. The results demonstrate that $k$-means acceleration yields substantial reductions in projection and attack time, while preserving attack success rates. These findings confirm that our approach makes PGD scalable to large discrete domains without sacrificing effectiveness, establishing both practical efficiency and rigorous error control. Our contributions are threefold: (1) We identify projection as the primary bottleneck of PGD in large discrete feasible sets. (2) We propose a $k$-means acceleration method for PGD with guaranteed cer-

tificates ensuring the validity of descent and convergence under inexact projections. (3) We provide theoretical guarantees and empirical validation, showing significant speedups without compromising adversarial success.

## 2 METHOD

We consider the constrained optimization problem

$$\min_{x \in \mathcal{C}} f(x), \qquad \mathcal{C} \subset \mathbb{R}^d, \tag{1}$$

where $f : \mathbb{R}^d \to \mathbb{R}$ is the loss function, and the set $\mathcal{C}$ encodes some structural constraints. For example, $\mathcal{C}$ is discrete for token embeddings in a language model vocabulary, codewords in quantization, or other large discrete dictionaries. A standard approach to this problem is projected gradient descent (PGD): starting from an initial example $x_0$, PGD alternates between an unconstrained gradient step in the continuous space and a projection back onto $\mathcal{C}$. Concretely, at iteration $t$, we form the intermediate (unprojected) point

$$u_t = x_t - \eta \nabla f(x_t). \tag{2}$$

and then enforce feasibility by projecting

$$x_{t+1} = \Pi_{\mathcal{C}}(u_t) := \arg \min_{z \in \mathcal{C}} \|z - u_t\|^2, \tag{3}$$

where $\| \cdot \|$ denotes the norm $l_2$. The two-step procedure is demonstrated in the Figure 1.

One adversarial attack proposed by Sadrizadeh et al. (2022) on LLMs follows exactly this PGD framework, adapted to the discrete embedding space: the objective $f$ measures classification (or generation) loss for the current input and $\mathcal{C}$ is the set of allowable token embeddings, so the algorithm seeks a feasible perturbation of $x_0$ that causes misclassification while preserving discreteness. In practice, the projection step is the computational bottleneck. To project an intermediate point $u_t$ back onto the feasible set $\mathcal{C}$, one must search over the entire vocabulary of size $V$ and compute the similarity (e.g., inner product or Euclidean distance) between $u_t$ and each candidate embedding. This requires $O(Vd)$ similarity evaluations in dimension $d$ at every iteration. For large language models, $V$ can be on the order of hundreds of thousands and $d$ may range from hundreds to thousands, this cost thus renders standard PGD infeasible at scale.

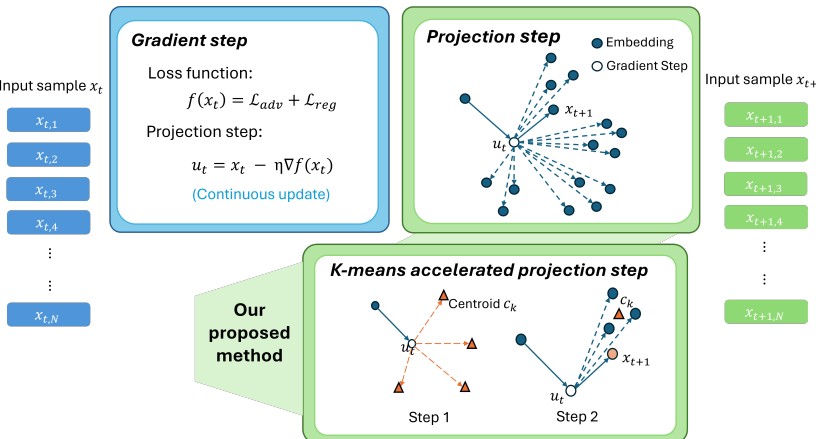

Figure 1: One iteration of projected gradient descent (PGD). **Top (standard PGD):** take a gradient step $u_t := x_t - \eta \nabla f(x_t)$, then project $u_t$ onto the discrete embedding set to obtain $x_{t+1}$. **Bottom (K-means–accelerated PGD, K-PGD):** replace the full projection with a two-stage $k$-means procedure—(1) score cluster centroids to build a shortlist, and (2) search only the shortlisted cluster(s) to project and produce $x_{t+1}$. Blue circles denote embeddings; the hollow circle denotes $u_t$; orange triangles denote centroids.

We aim to accelerate the projection step while preserving the effectiveness of PGD and significantly improving its efficiency. The key idea is to approximate the projection by *clustering the feasible*

*set* using spherical $k$-means. The method consists of two components: (1) an offline preprocessing stage that clusters that vocabulary embeddings using iterative $k$-means, and (2) an online projection step that leverages these clusters to accelerate PGD.

---

**Algorithm 1** Iterative K-means Preprocessing (IKMP)

---

**Require:** maxIter: maximum iteration number; $|\mathcal{V}|$: number of embeddings; $K$: number of clusters; tolerance $\epsilon > 0$

**Ensure:** centroids $\{c_1, \ldots, c_K\}$, clusters $\{\mathcal{V}_1, \ldots, \mathcal{V}_K\}$

1: Initialize $K$ centroids $\{c_k^{(0)}\}$ randomly from $\mathcal{V}$
2: iter $\leftarrow 0$
3: **while** iter $<$ maxIter **do**
4:      Reset all buckets $\{\mathcal{V}_k\}_{k=1}^{K}$ to empty
5:      **for** each embedding $e_i \in \mathcal{V}$ **do**
6:          Assign $e_i$ to the nearest centroid by cosine similarity
7:          Add $e_i$ to the corresponding bucket
8:      **end for**
9:      Update each centroid $\hat{c}_k^{(t+1)} = \frac{1}{|\mathcal{V}_k^{(t)}|} \sum_{e_i \in \mathcal{V}_k^{(t)}} e_i$
10:     Normalize each centroid $c_k^{(t+1)} = \frac{\hat{c}_k^{(t+1)}}{\|\hat{c}_k^{(t+1)}\|}$
11:     iter $\leftarrow$ iter + 1
12:     **if** $\max_{1 \leq k \leq K} \|c_k^{(t+1)} - c_k^{(t)}\|_2 \leq \epsilon$ **then**
13:         **break**
14:     **end if**
15: **end while**
16: **return** final centroids $\{c_k\}$ and clusters $\{\mathcal{V}_k\}$

---

**Iterative K-means Preprocessing.** The preprocessing stage partitions the vocabulary embeddings into $K$ coherent clusters using an iterative $K$-means procedure (Algorithm 1). Let $\mathcal{V} = \{e_1, e_2, \ldots, e_V\}$ denote the set of token embeddings, each normalized to unit length ($\|e_v\| = 1$). We begin by selecting $K$ embeddings from $\mathcal{V}$ randomly to serve as initial centroids $\{c_1^{(0)}, \ldots, c_K^{(0)}\}$ (Line 1). At iteration $t$, each embedding $e_i$ is assigned to the nearest centoird according to cosine similarity:

$$\text{assign}(e_i) = \arg \max_{1 \leq k \leq K} \langle e_i, c_k^{(t)} \rangle.$$

All embeddings assigned to centroid $c_k^{(t)}$ form the cluster $\mathcal{V}_k^{(t)}$ (Line 4-7). Each centroid is then updated as the arithmetic mean of its assigned embeddings (Line 9):

$$\hat{c}_k^{(t+1)} = \frac{1}{\|\mathcal{V}_k^{(t)}\|} \sum_{e_i \in \mathcal{V}_k^{(t)}} e_i.$$

To ensure consistency with the unit-norm embeddings, the updated centroid is projected back to the unit sphere (Line 10):

$$c_k^{(t+1)} = \frac{\hat{c}_k^{(t+1)}}{\|\hat{c}_k^{(t+1)}\|}.$$

Normalization is crucial for two reasons. First, it guarantees the centroid-embedding comparison remain meaningful under cosine similarity; Second, it ensures that the cluster radius, defined as $R_k = \max_{e_i \in \mathcal{V}_k} \|e_i - c_k\|$, reflects purely angular deviation within the cluster rather than magnitude differences. The procedure repeats until either convergence is reached or the maximum number of iterations is met (Lines 11–13). The final set of normalized centroids $\{c_k\}$ along with their corresponding clusters $\{\mathcal{V}_k\}$ is returned (Line 16).

This clustering step is performed once as a preprocessing stage and is amortized across all PGD iterations. By structuring the vocabulary into centroid-based groups, we enable the efficient shortlist-based projection used in K-means–Accelerated PGD (Algorithm 2).

---

**Algorithm 2** K-Means–Accelerated PGD (K-PGD)

---

**Require:** objective $f$, step size $\eta > 0$, max iters $K$, centroids $\{c_k\}_{k=1}^K$, clusters $\{\mathcal{V}_k\}_{k=1}^K$, shortlist size $M$

**Ensure:** final iterate $x_T$

1: initialize $x_0$
2: **for** $t = 0, 1, \ldots, K-1$ **do**
3:     **Gradient step:**
$$u_t \leftarrow x_t - \eta \nabla f(x_t), \qquad \tilde{u}_t \leftarrow \frac{u_t}{\|u_t\|}$$

4:     **Centroid scoring:**
$$s_k \leftarrow \langle \tilde{u}_t, c_k \rangle \quad \text{for } k = 1, \ldots, K$$

5:     **Select top-$M$ clusters:**
$$S_t \leftarrow \mathrm{TopM}\big(\{s_k\}_{k=1}^K, \ M\big)$$

6:     **Restricted projection:**
$$x_{t+1} \leftarrow \arg \max_{e \in \cup_{j \in S_t} \mathcal{V}_j} \langle \tilde{u}_t, e \rangle$$

7: **end for**
8: **return** $x_{t+1}$

---

**K-means-Accelerated PGD (K-PGD).** The K-PGD procedure accelerates projected gradient descent by restricting the projection step to a shortlist of candidate clusters.

The algorithm begins by initializing the input example $x_0$ (Line 1). At each iteration $t$, a gradient descent step is performed in the continuous embedding space:

$$u_t = x_t - \eta \nabla f(x_t), \qquad \tilde{u}_t = \frac{u_t}{\|u_t\|},$$

where $u_t$ is the unconstrained update and $\tilde{u}_t$ is normalized to lie on the unit sphere for cosine-based comparisons (Line 3).

Next, similarity scores are computed between $\tilde{u}_t$ and all cluster centroids $\{c_k\}_{k=1}^K$:

$$s_k = \langle \tilde{u}_t, c_k \rangle, \quad k = 1, \ldots, K,$$

providing a measure of alignment between the update direction and each cluster (Line 4). Based on these scores, the algorithm selects the top-$M$ clusters with highest similarity values, forming a shortlist $S_t$ (Line 5).

Finally, the projection step is restricted to the embeddings in the shortlisted clusters:

$$x_{t+1} = \arg \max_{e \in \cup_{j \in S_t} \mathcal{V}_j} \langle \tilde{u}_t, e \rangle,$$

which identifies the nearest embedding only among the candidates in the selected clusters (Line 6). This reduces the per-iteration cost from $O(Vd)$ to $O(Kd + (MV/K)d)$, where the first term arises from centroid scoring and the second term corresponds to the restricted exact search.

The procedure repeats for at most $K$ iterations or until an adversarial example is found. The final iterate $x_T$ is then returned as output (Lines 8).

**Complexity.** Each iteration of the accelerated method requires computing $K$ inner products with centroids, selecting the top-$M$, and scanning $\sum_{j \in S_t} |\mathcal{V}_j| \approx (MV/K)$ candidates. The resulting complexity is $O(Kd + (MV/K)d)$ per iteration, compared to $O(Vd)$ for the standard PGD projection. For typical settings $M \ll K \ll V$, this yields more than an order-of-magnitude speedup without harming convergence.

## 3 THEORETICAL ANALYSIS

In this section, we establish formal guarantees for our approximate projection framework. Section 3.2 develops the theoretical foundation by introducing score certificates and proving that they imply

Q-certificates, which control the projection error in terms of the $\delta$-proximal condition. This provides an abstract guarantee: as long as the score gap between missed and kept candidates is bounded, the approximate projection enjoys the same convergence properties as the exact projection. Section 3.3 then instantiates this guarantee by constructing deterministic certificates from spherical $k$-means envelopes. Here, centroids and cluster radii are used to derive computable bounds on inner products, yielding practical conditions under which the theoretical guarantees hold. Together, these results show that clustering-based shortlists not only accelerate projection steps but also admit provable robustness guarantees.

### 3.1 General convergence guarantees under $\delta$-proximal projections

Before we present our theoretical results, we make the following assumptions on the loss function $f$ in equation 2 and the feasible set $\mathcal{C}$ in the optimization equation 3.

**Assumption 1** (Smoothness). *The objective $f : \mathbb{R}^d \to \mathbb{R}$ is differentiable and $L$-smooth, i.e.,*

$$\|\nabla f(x) - \nabla f(y)\| \le L\|x - y\|, \qquad \forall x, y \in \mathbb{R}^d.$$

**Assumption 2** (Feasible set). *The feasible region $\mathcal{C} \subset \mathbb{R}^d$ is nonempty. We allow $\mathcal{C}$ to be nonconvex or even discrete (e.g., a large vocabulary in a language model).*

When applying the K-means acceleration, PGD returns an approximate Projection $\hat{x}_{t+1}$ as the minimizer of $\|z - u_t\|^2$ within cluster shortlist $S_t$. Hereby, we introduce the notion of a $\delta$-*proximal certificate*. This definition captures the idea that the approximate projection need not be exact, but should be close enough to the true projection in terms of the proximal surrogate objective.

**Definition 1** ($\delta$-proximal certificate). *A point $\hat{x}_{t+1} \in \mathcal{C}$ satisfies a $\delta_t$-proximal certificate at $u_t$ if*

$$\frac{1}{2\eta}\|\hat{x}_{t+1} - u_t\|^2 \ \le \ \min_{z \in \mathcal{C}} \frac{1}{2\eta}\|z - u_t\|^2 + \delta_t. \tag{4}$$

Formally, the certificate quantifies how far $\hat{x}_{t+1}$ is from the exact solution of the projection subproblem to be cleanly propagated into a convergence guarantee. Thus, the $\delta$-prox condition allows us to bridge between approximate projection and rigorous optimization theory. Equivalently, in terms of the proximal surrogate

$$Q(z; x_t) = f(x_t) + \langle \nabla f(x_t), z - x_t \rangle + \tfrac{1}{2\eta}\|z - x_t\|^2,$$

condition equation 4 means

$$Q(\hat{x}_{t+1}; x_t) \le \min_{z \in \mathcal{C}} Q(z; x_t) + \delta_t.$$

Once we have defined $\delta$-proximal certificates, the next step is to understand how they affect the progress of the PGD iteration. The standard PGD analysis relies on the fact that the projection enforces a decrease in a local surrogate of the objective. With approximate projections, this decrease may not hold exactly, but we can show that a relaxed version still applies, which is formally stated in the following Lemma.

**Lemma 1** (Inexact PGD descent, proved in Appendix A.1). *Suppose $f$ is $L$-smooth and $\eta \le 1/L$. If $\hat{x}_{t+1}$ satisfies a $\delta_t$-proximal certificate at $u_t$, then*

$$f(\hat{x}_{t+1}) \ \le \ f(x_t) - \left(\tfrac{1}{2\eta} - \tfrac{L}{2}\right)\|\hat{x}_{t+1} - x_t\|^2 + \delta_t.$$

Lemma 1 establishes that each update decreases the function value up to a small additive term $\delta_t$, which directly reflects the inexactness of the projection. This lemma is the key technical stepping stone: it ensures that even though we project approximately, the algorithm still makes descent progress. Without such a result, there would be no way to propagate the approximation error into a global convergence theorem.

**Theorem 1** (Convergence with $\delta$-prox certificates, proved in Appendix A.2). *Let $\bar{\delta} = \frac{1}{T}\sum_{t=0}^{T-1} \delta_t$ be the average projection error. Assume $f$ is $L$-smooth, $\eta \le 1/L$, and each step satisfies equation 4. Then for all $T \ge 1$:*

1. **General $\mathcal{C}$ (possibly nonconvex):**

$$\min_{0 \leq t < T} \|\hat{x}_{t+1} - x_t\|^2 \leq \frac{2\eta}{1 - \eta L}\left(\frac{f(x_0) - f^\star}{T} + \bar{\delta}\right).$$

2. **Convex $\mathcal{C}$:** *Let $G_\eta(x_t) = \frac{1}{\eta}(x_t - \Pi_\mathcal{C}(x_t - \eta\nabla f(x_t)))$. Then*

$$\min_{0 \leq t < T} \|G_\eta(x_t)\|^2 \leq \frac{2(f(x_0) - f^\star)}{\eta T} + \frac{2}{\eta}\bar{\delta}.$$

Finally, Theorem 1 builds on the descent lemma to show global convergence guarantees for the entire PGD trajectory. By summing the descent inequality across iterations, we can control the average stationarity gap in terms of both the iteration count $T$ and the average projection error $\bar{\delta}$. This yields a natural trade-off: faster but less precise projections (i.e., larger $\bar{\delta}$) still give convergence, but to a neighborhood whose size is governed by $\bar{\delta}$. Exact projections correspond to $\bar{\delta} = 0$, recovering the classical PGD guarantee.

### 3.2 How the $k$-means envelope yields Q-certificates

To certify the quality of approximate projections, we need a way to reason about all atoms inside a cluster without checking them individually. The $k$-means *envelope* provides exactly this: each cluster $\mathcal{V}_k$ is summarized by a centroid $c_k$ and a radius $R_k = \max_{v \in \mathcal{V}_k} \|e_v - c_k\|$, so that every atom $e \in \mathcal{V}_k$ lies within $R_k$ of its centroid. This construction (Figure 2) allows us to bound the inner product $\langle \tilde{u}_t, e \rangle$ with any atom $e$ in the cluster using only the centroid and radius, instead of enumerating all elements of $\mathcal{V}_k$. Such bounds are crucial because they let us control the error incurred when restricting the search to a shortlist of clusters.

**Envelope (centroid+radius) bounds.** For any $e \in \mathcal{V}_k$ we have

$$\langle \tilde{u}_t, e \rangle = \langle \tilde{u}_t, c_k \rangle + \langle \tilde{u}_t, e - c_k \rangle \in \left[\langle \tilde{u}_t, c_k \rangle - \|e - c_k\|, \ \langle \tilde{u}_t, c_k \rangle + \|e - c_k\|\right] \subseteq \left[\langle \tilde{u}_t, c_k \rangle - R_k, \ \langle \tilde{u}_t, c_k \rangle + R_k\right]. \tag{5}$$

Hence the *best missed score* and the *best kept score* obey

$$U_{\text{miss}} := \max_{k \notin S_t} \max_{e \in \mathcal{V}_k} \langle \tilde{u}_t, e \rangle \leq \max_{k \notin S_t}\left(\langle \tilde{u}_t, c_k \rangle + R_k\right), \tag{6}$$

$$L_{\text{keep}} := \max_{j \in S_t} \max_{e \in \mathcal{V}_j} \langle \tilde{u}_t, e \rangle \geq \max_{j \in S_t}\left(\langle \tilde{u}_t, c_j \rangle - R_j\right). \tag{7}$$

Define the *score certificate*

$$\varepsilon_t^{\text{cert}} := \left[U_{\text{miss}} - L_{\text{keep}}\right]_+ = \max\left\{\max_{k \notin S_t}\left(\langle \tilde{u}_t, c_k \rangle + R_k\right) - \max_{j \in S_t}\left(\langle \tilde{u}_t, c_j \rangle - R_j\right), \ 0\right\}. \tag{8}$$

**Lemma 2** (Envelope $\Rightarrow$ score gap certificate, proved in Appendix A.1)**.** *Let $e^\star \in \arg\max_{e \in \mathcal{C}} \langle \tilde{u}_t, e \rangle$ and let $\widehat{e}_t$ be the best element found inside the shortlist. Then*

$$\langle \tilde{u}_t, e^\star \rangle - \langle \tilde{u}_t, \widehat{e}_t \rangle \leq \varepsilon_t^{\text{cert}}.$$

Lemma 2 ensures that the best missed atom cannot outperform the best kept atom by more than $\varepsilon_t^{\text{cert}}$. This turns per-cluster envelope bounds into a global guarantee on the quality of the shortlist. The next step is to show that such a score gap directly controls the $\delta$-prox error in the projection subproblem.

**Theorem 2** (Score certificate $\Rightarrow$ Q-certificate, proved in Appendix A.3)**.** *Assume $\|e\| = 1$ for all $e \in \mathcal{C}$. Let $e^\star \in \arg\max_{e \in \mathcal{C}} \langle u_t, e \rangle$ (equivalently the exact projection of $u_t$ onto $\mathcal{C}$), and let $\widehat{e}_t$ be the shortlist maximizer. Then the approximate projection $\hat{x}_{t+1} := \widehat{e}_t$ satisfies the $\delta$-prox condition equation 4 with*

$$\delta_t = \frac{\|u_t\|}{\eta}\varepsilon_t^{\text{cert}}. \tag{9}$$

Theorem 2 upgrades the score gap guarantee into a full Q-certificate: if the shortlist preserves near-optimality in terms of inner products, then the resulting approximate projection enjoys a bounded $\delta$-prox error. This step bridges local cluster-based control to the global convergence analysis in the next section.

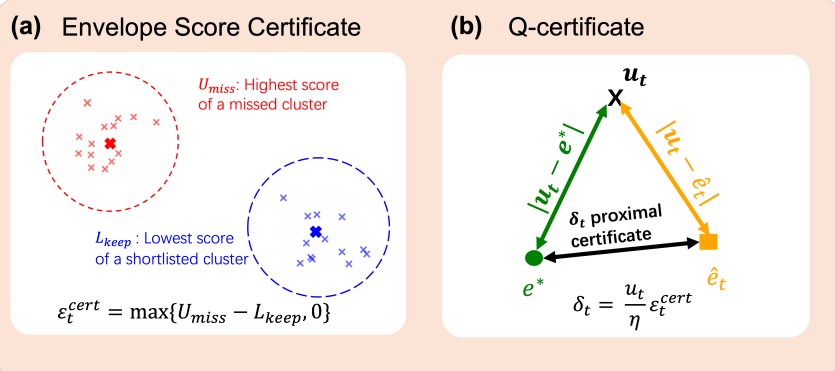

Figure 2: (a) The score gap is the difference between the best possible score from a missed cluster and the best guaranteed score from a kept cluster. (b) The exact projection $e^*$ (green) and approximate projection $\hat{e}_t$ (orange) yield different distances to the gradient step $u_t$ (black). Then the score certificate implies the Q-certificate, where the inner product gap between $e^\star$ and $\hat{e}_t$ is bounded by $\varepsilon_t^{\text{cert}}$.

### 3.3 CONVERGENCE TIME WITH ACCUMULATED CERTIFICATE ERROR

We now establish convergence guarantees for our method under accumulated certificate error. The results show that approximate projections guided by certificates achieve the same asymptotic rate as exact PGD, up to an additive error floor.

**Theorem 3** (Convergence with cluster-based certificates, proved in Appendix A.4). *Suppose $f$ is $L$-smooth and $\eta \leq 1/L$. At each iteration, the approximate projection is computed using the $k$-means shortlist with envelope bounds. Then after $T$ iterations,*

$$\min_{0 \leq t < T} \|\hat{x}_{t+1} - x_t\|^2 \leq \frac{2\eta}{1 - \eta L}\left(\frac{f(x_0) - f^\star}{T} + \bar{\delta}_T\right), \quad \bar{\delta}_T = \frac{1}{T}\sum_{t=0}^{T-1}\delta_t, \quad \delta_t = \frac{\|u_t\|}{\eta}\varepsilon_t^{cert}.$$

Theorem 3 shows that accelerated PGD converges at the same $O(1/T)$ rate as exact PGD, up to an additive error floor determined by the average certificate gap. If the certificate error vanishes ($\varepsilon_t^{\text{cert}} = 0$), we recover the standard PGD guarantee.

**Theorem 4** (Convergence time with accumulated error, proved in Appendix A.5). *Assume $\eta \leq 1/L$ and let $\bar{\Delta}$ be any bound such that $\bar{\delta}_T \leq \bar{\Delta}$ for all $T$. If*

$$\varepsilon^2 > \frac{2\eta}{1 - \eta L}\bar{\Delta},$$

*then the number of iterations required to reach accuracy $\varepsilon$ is bounded by*

$$T \geq \frac{f(x_0) - f^\star}{\frac{1}{1-\eta L}\frac{\varepsilon^2}{2\eta} - \bar{\Delta}}.$$

Theorem 4 refines Theorem 3 by quantifying the iteration complexity in terms of $\varepsilon$. The bound highlights the role of $\bar{\Delta}$ as a *floor of approximation*: when $\bar{\Delta}$ is small, the usual $O(1/T)$ behavior dominates, while nonzero $\bar{\Delta}$ slows progress but still ensures convergence to an $\varepsilon$-ball.

## 4 EXPERIMENTS AND RESULTS

**Settings and Datasets.** We evaluate both the original PGD attack and the IKMP-Accelerated PGD attack to GPT-2 model in text classification task on three widely used benchmark datasets: Yelp Reviews Datasets (2025), IMDB by Maas et al. (2011), and SNLI Bowman et al. (2015). The Yelp Review dataset contains user-generated business reviews with associated star ratings, providing rich sentiment-oriented text commonly used for sentiment classification tasks. The IMDB dataset

| Algorithm | Initial $K$ | Token Error Rate (TER) | Avg Cosine Sim (ACS) | Success Attack Rate (SAR) | Attack Time |
|---|---|---|---|---|---|
| PGD | 10 | $0.078 \pm 0.012$ | $0.423 \pm 0.0123$ | $0.615 \pm 0.017$ | $0.145 \pm 0.0337$ |
| IKMP + PGD | 10 | $0.182 \pm 0.0097 \uparrow$ | $0.487 \pm 0.0092 \uparrow$ | $0.855 \pm 0.023 \uparrow$ | $0.070 \pm 0.039 \downarrow$ |
| PGD | 11 | $0.086 \pm 0.013$ | $0.427 \pm 0.0135$ | $0.620 \pm 0.021$ | $0.705 \pm 0.0624$ |
| IKMP + PGD | 11 | $0.184 \pm 0.0083 \uparrow$ | $0.496 \pm 0.0101 \uparrow$ | $0.865 \pm 0.012 \uparrow$ | $0.110 \pm 0.031 \downarrow$ |
| PGD | 12 | $0.115 \pm 0.008$ | $0.432 \pm 0.00921$ | $0.660 \pm 0.020$ | $0.669 \pm 0.0529$ |
| IKMP + PGD | 12 | $0.166 \pm 0.0132 \uparrow$ | $0.495 \pm 0.0122 \uparrow$ | $0.850 \pm 0.014 \uparrow$ | $0.100 \pm 0.0219 \downarrow$ |
| PGD | 13 | $0.175 \pm 0.013$ | $0.446 \pm 0.0118$ | $0.720 \pm 0.018$ | $0.108 \pm 0.0428$ |
| IKMP + PGD | 13 | $0.162 \pm 0.0082 \downarrow$ | $0.506 \pm 0.0092 \uparrow$ | $0.880 \pm 0.015 \uparrow$ | $0.227 \pm 0.031 \uparrow$ |
| PGD | 14 | $0.117 \pm 0.0074$ | $0.441 \pm 0.0123$ | $0.665 \pm 0.017$ | $0.093 \pm 0.0418$ |
| IKMP + PGD | 14 | $0.158 \pm 0.00926 \uparrow$ | $0.515 \pm 0.0101 \uparrow$ | $0.890 \pm 0.018 \uparrow$ | $0.094 \pm 0.0421 \uparrow$ |
| PGD | 15 | $0.223 \pm 0.009$ | $0.471 \pm 0.0107$ | $0.815 \pm 0.020$ | $0.690 \pm 0.0547$ |
| IKMP + PGD | 15 | $0.147 \pm 0.0083 \downarrow$ | $0.535 \pm 0.0106 \uparrow$ | $0.915 \pm 0.019 \uparrow$ | $0.536 \pm 0.0322 \downarrow$ |
| PGD | 16 | $0.219 \pm 0.005$ | $0.473 \pm 0.0108$ | $0.840 \pm 0.018$ | $0.177 \pm 0.0842$ |
| IKMP + PGD | 16 | $0.154 \pm 0.0092 \downarrow$ | $0.538 \pm 0.0091 \uparrow$ | $0.935 \pm 0.021 \uparrow$ | $0.750 \pm 0.0428 \uparrow$ |
| PGD | 17 | $0.229 \pm 0.002$ | $0.491 \pm 0.0121$ | $0.890 \pm 0.016$ | $0.490 \pm 0.0391$ |
| IKMP + PGD | 17 | $0.164 \pm 0.00831 \downarrow$ | $0.534 \pm 0.0127 \uparrow$ | $0.935 \pm 0.021 \uparrow$ | $0.102 \pm 0.0391 \downarrow$ |
| PGD | 18 | $0.213 \pm 0.003$ | $0.506 \pm 0.0116$ | $0.905 \pm 0.022$ | $0.374 \pm 0.0284$ |
| IKMP + PGD | 18 | $0.154 \pm 0.0104 \downarrow$ | $0.540 \pm 0.0130 \uparrow$ | $0.940 \pm 0.018 \uparrow$ | $0.354 \pm 0.0303 \downarrow$ |
| PGD | 19 | $0.210 \pm 0.0065$ | $0.516 \pm 0.0130$ | $0.910 \pm 0.009$ | $0.260 \pm 0.0472$ |
| IKMP + PGD | 19 | $0.155 \pm 0.0091 \downarrow$ | $0.546 \pm 0.0116 \uparrow$ | $0.965 \pm 0.017 \uparrow$ | $0.562 \pm 0.045 \uparrow$ |
| PGD | 20 | $0.189 \pm 0.0048$ | $0.522 \pm 0.0134$ | $0.930 \pm 0.013$ | $1.135 \pm 0.0328$ |
| IKMP + PGD | 20 | $0.140 \pm 0.0107 \downarrow$ | $0.555 \pm 0.0123 \uparrow$ | $0.965 \pm 0.012 \uparrow$ | $1.109 \pm 0.0425 \downarrow$ |

Table 1: Results of PGD and IKMP+PGD attacks for $K = 10$ to 20. The table reports token error rate (TER), average cosine similarity (ACS), success attack rate (SAR), and attack time. For each $K$, the IKMP+PGD entry includes an arrow indicating its trend relative to PGD at the same $K$ ($\uparrow$ greater, $\downarrow$ smaller).

consists of 50,000 movie reviews labeled for binary sentiment (positive vs. negative), balanced across training and test splits, making it a standard benchmark for sentiment analysis and adversarial robustness evaluation. The Stanford Natural Language Inference (SNLI) dataset includes 570,000 human-annotated sentence pairs labeled as entailment, contradiction, or neutral, and serves as a key benchmark for natural language inference under adversarial perturbations.

We evaluate our proposed method under a range of hyperparameter and initialization settings. The initial number of clusters is denoted as $2^K$. The hyperparameter $\alpha$ controls the step size scaling in the projected gradient updates, balancing progress along the adversarial direction with stability of the update. We evaluate $\alpha \in \{10, 8, 5, 2\}$ in all experiments. To ensure experimental robustness across domains, we run experiments on three datasets: (1) a subset of 1,000 sentences from the YELP Review corpus (sentiment domain), (2) 1,000 sentences from the AG News dataset (topic classification), and (3) 1,000 sentences from the IMDB dataset (longer-form reviews). For all datasets, we report sentence-level results with $K$ varying from 10 to 20 in increments of 1. For every $K$ we conduct 10 independent, randomized trials.

**Evaluation metrics.** We compare the proposed IKMP+PGD attack against the standard PGD attack baseline using the following metrics: (i) **Attack Time** $T = \frac{\text{Total runtime}}{\text{Number of iterations}}$. The average runtime per PGD iteration, measured in seconds and calculated as (ii) **Average Cosine Similarity** $\text{ACS} = \frac{1}{N} \sum_{i=1}^{N} \frac{\langle x_i, v_i \rangle}{\|x_i\| \|v_i\|}$. The average cosine similarity between the attacked embedding $x_i$ and the clean embedding $v_i$. (iii) **Successful Attack Rate** $\text{SAR} = \frac{\#\{\text{successful attacks}\}}{\#\{\text{total runs}\}}$. The fraction of runs in which accelerated PGD successfully finds an adversarial example. (iv) **Token Error Rate** $\text{TER} = \frac{\#\text{Tokens changed}}{\#\text{Tokens}}$. The proportion of tokens in the original sequence that are modified by the attack, thereby measuring the overall extent of perturbation. For each metric (TER, ACS, SAR, and attack time), we highlight improvements relative to PGD. Entries with smaller TER, larger ACS, larger SAR, or reduced attack time are marked in blue.

**Results.** Overall, our results show that IKMP+PGD achieves consistently higher ACS, reduced TER, and improved SAR compared to standard PGD, while also reducing average attack time sig-

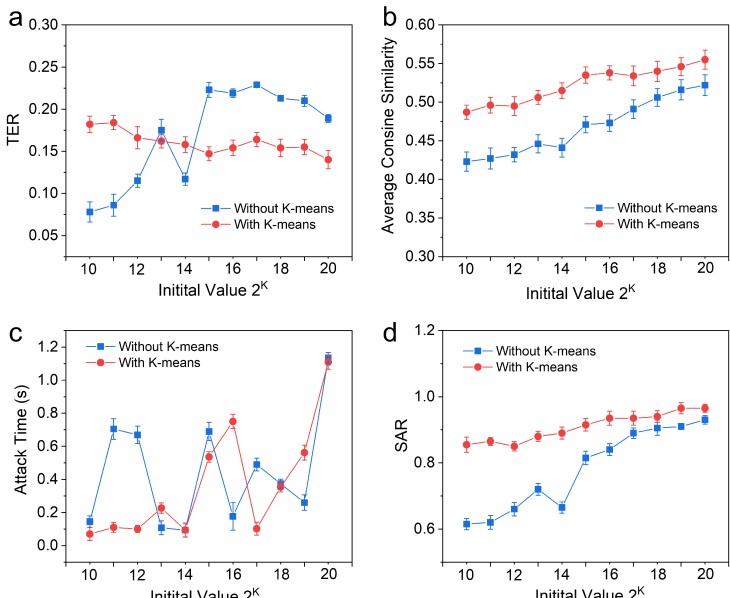

Figure 3: Evaluation of IKMP+PGD (with K-means, red) and standard PGD (without K-means, blue) across different initial values $2^K$. (a) TER, (b) ACS, (c) Attack Time per sample (s), and (d) SAR. Each curve shows the mean value across runs, with error bars denoting the standard deviation.

nificantly. This demonstrates that clustering-based initialization provides both efficiency and robustness gains. Detailed numerical comparisons are presented in Tables 1. The results in Figure 3 demonstrate consistent improvements when K-means clustering is integrated into PGD. In subplot (a), token error rate (TER) is substantially lower with K-means across most initial values, showing greater stability and less variance. Subplot (b) shows that average cosine similarity (ACS) is consistently higher with K-means, indicating that updates are more aligned with the clean embeddings. In subplot (c), attack time is generally reduced or comparable when using K-means, with especially noticeable improvements at lower values of $K$. Finally, subplot (d) illustrates that the successful attack rate (SAR) is consistently higher for K-means across all initializations, confirming the robustness advantage of clustering-based initialization. Collectively, these plots indicate that the proposed IKMP+PGD approach yields better performance, efficiency, and stability than standard PGD.

## 5 DISCUSSION

We proposed K-PGD, a clustering-augmented variant of projected gradient descent, which accelerates adversarial search through approximate projection while preserving convergence properties. Theoretical analysis established error accumulation bounds and introduced the Q-certificate assumption, providing conditions under which robustness guarantees hold. Empirically, evaluations on Yelp, IMDB, and SNLI demonstrated consistent reductions in Token Error Rate and improvements in Average Cosine Similarity and Success Attack Rate, confirming both the efficiency and effectiveness of the proposed framework. A limitation of our approach is that the runtime improvements are not uniform across all cluster sizes. For certain number of clusters, the overhead of managing clusters offsets the pruning benefits, leading to no reduction in attack time. This highlights the sensitivity of efficiency gains to the choice of $K$.

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
