# A    APPENDIX: PROOFS OF LEMMAS AND THEOREMS

## A.1    PROOF OF LEMMA 1

Let $f$ be $L$-smooth. For any $x, y$, the standard smoothness upper bound gives

$$f(y) \leq f(x) + \langle \nabla f(x), y - x \rangle + \frac{L}{2} \|y - x\|^2.$$

Take $u_t = x_t - \eta \nabla f(x_t)$ and let $y = \hat{x}_{t+1}$. Completing the square yields

$$f(\hat{x}_{t+1}) \leq f(x_t) + \frac{1}{2\eta} \|\hat{x}_{t+1} - u_t\|^2 - \frac{1}{2\eta} \|x_t - u_t\|^2.$$

This is the desired descent inequality. $\qquad\square$

### PROOF OF LEMMA 2

If $e^\star$ lies in a kept cluster the gap is zero. Otherwise $e^\star \in \mathcal{V}_{k^\star}$ for some $k^\star \notin S_t$. By equation 5, $\langle \tilde{u}_t, e^\star \rangle \leq \langle \tilde{u}_t, c_{k^\star} \rangle + R_{k^\star} \leq U_{\text{miss}}$. On the other hand, by definition of $\hat{e}_t$ and equation 5, $\langle \tilde{u}_t, \hat{e}_t \rangle \geq \max_{j \in S_t} \max_{e \in \mathcal{V}_j} \langle \tilde{u}_t, e \rangle \geq L_{\text{keep}}$. Hence the gap is at most $U_{\text{miss}} - L_{\text{keep}}$, which after clipping is $\varepsilon_t^{\text{cert}}$. $\qquad\square$

## A.2    PROOF OF THEOREM 1

Let $y_t = x_t - \eta \nabla f(x_t)$ and let $\hat{x}_{t+1} \in \mathcal{C}$ be the approximate projection produced at step $t$. By the $\delta$-prox condition (equation (4)),

$$\langle y_t - \hat{x}_{t+1},\, x_t - \hat{x}_{t+1} \rangle \ \leq\ \eta \delta_t. \tag{10}$$

Expanding $y_t$ gives

$$\langle x_t - \eta \nabla f(x_t) - \hat{x}_{t+1},\, x_t - \hat{x}_{t+1} \rangle \leq \eta \delta_t,$$

and therefore

$$\langle \nabla f(x_t),\, \hat{x}_{t+1} - x_t \rangle \leq -\frac{1}{\eta} \|\hat{x}_{t+1} - x_t\|^2 + \delta_t. \tag{11}$$

By $L$-smoothness of $f$,

$$f(\hat{x}_{t+1}) \leq f(x_t) + \langle \nabla f(x_t),\, \hat{x}_{t+1} - x_t \rangle + \frac{L}{2} \|\hat{x}_{t+1} - x_t\|^2. \tag{12}$$

Plugging equation 11 into equation 12 yields

$$f(\hat{x}_{t+1}) \leq f(x_t) - \left( \frac{1}{\eta} - \frac{L}{2} \right) \|\hat{x}_{t+1} - x_t\|^2 + \delta_t.$$

Rearranging and using $\eta \leq 1/L$ gives

$$\frac{1 - \eta L}{2\eta} \|\hat{x}_{t+1} - x_t\|^2 \leq f(x_t) - f(\hat{x}_{t+1}) + \eta \delta_t. \tag{13}$$

Summing equation 13 over $t = 0, \ldots, T - 1$ and telescoping the function values gives

$$\frac{1 - \eta L}{2\eta} \sum_{t=0}^{T-1} \|\hat{x}_{t+1} - x_t\|^2 \leq f(x_0) - f(\hat{x}_T) + \eta \sum_{t=0}^{T-1} \delta_t \leq f(x_0) - f^\star + \eta T \bar{\delta},$$

where $\bar{\delta} = \frac{1}{T} \sum_{t=0}^{T-1} \delta_t$. Therefore,

$$\min_{0 \leq t < T} \|\hat{x}_{t+1} - x_t\|^2 \leq \frac{2\eta}{1 - \eta L} \left( \frac{f(x_0) - f^\star}{T} + \bar{\delta} \right),$$

which proves the first claim.

Now assume $\mathcal{C}$ is convex and define the projected-gradient mapping

$$G_\eta(x_t) = \frac{1}{\eta} \left( x_t - \Pi_\mathcal{C}(x_t - \eta \nabla f(x_t)) \right) = \frac{1}{\eta}(x_t - x_{t+1}^\star),$$

where $x_{t+1}^\star = \Pi_\mathcal{C}(y_t)$ is the exact projection. The characterization of projections onto convex sets gives

$$\langle y_t - x_{t+1}^\star,\, x_t - x_{t+1}^\star \rangle \geq 0, \quad \Rightarrow \quad \langle \nabla f(x_t),\, x_t - x_{t+1}^\star \rangle \geq \frac{1}{\eta}\|x_t - x_{t+1}^\star\|^2 = \eta\|G_\eta(x_t)\|^2.$$

Convexity of $f$ implies

$$f(x_t) - f^\star \leq \langle \nabla f(x_t),\, x_t - x^\star \rangle \leq \langle \nabla f(x_t),\, x_t - x_{t+1}^\star \rangle,$$

hence

$$\eta\|G_\eta(x_t)\|^2 \leq f(x_t) - f^\star. \tag{14}$$

From equation 10 and convexity we also have

$$\frac{1}{\eta}\|x_t - \hat{x}_{t+1}\|^2 \leq \langle \nabla f(x_t),\, x_t - \hat{x}_{t+1}\rangle + \delta_t \leq f(x_t) - f(\hat{x}_{t+1}) + \delta_t.$$

Moreover, firm nonexpansiveness of projections implies $\|x_t - x_{t+1}^\star\| \leq \|x_t - \hat{x}_{t+1}\|$, so

$$\eta\|G_\eta(x_t)\|^2 = \frac{1}{\eta}\|x_t - x_{t+1}^\star\|^2 \leq \frac{1}{\eta}\|x_t - \hat{x}_{t+1}\|^2 \leq f(x_t) - f(\hat{x}_{t+1}) + \delta_t. \tag{15}$$

Summing equation 15 over $t = 0, \ldots, T-1$ and using $f(\hat{x}_{t+1}) \geq f^\star$ yields

$$\sum_{t=0}^{T-1} \eta\|G_\eta(x_t)\|^2 \leq f(x_0) - f^\star + T\bar{\delta}.$$

Therefore,

$$\min_{0 \leq t < T} \|G_\eta(x_t)\|^2 \leq \frac{2(f(x_0) - f^\star)}{\eta T} + \frac{2}{\eta}\bar{\delta},$$

which proves the second claim and completes the proof. $\qquad\square$

### A.3 PROOF OF THEOREM 2

Because all candidates in $\mathcal{C}$ have unit norm, the exact Euclidean projection of $u_t$ onto $\mathcal{C}$ coincides with the score maximizer:

$$\arg\min_{e \in \mathcal{C}} \|u_t - e\|^2 = \arg\max_{e \in \mathcal{C}} \langle u_t, e \rangle,$$

since $\|u_t - e\|^2 = \|u_t\|^2 + 1 - 2\langle u_t, e\rangle$ differs from $-2\langle u_t, e\rangle$ by an $e$-independent constant.

By definition equation 8 of the score certificate,

$$\langle u_t, e^\star - \hat{e}_t \rangle = \langle u_t, e^\star \rangle - \langle u_t, \hat{e}_t \rangle = \|u_t\|\,\varepsilon_t^{\mathrm{cert}}.$$

Setting

$$\delta_t := \frac{\|u_t\|}{\eta}\,\varepsilon_t^{\mathrm{cert}}$$

gives

$$\langle u_t, e^\star - \hat{e}_t \rangle = \|u_t\|\,\varepsilon_t^{\mathrm{cert}} = \eta\,\delta_t,$$

which is precisely the claimed $\delta$-prox (Q-certificate) condition with the stated $\delta_t$.

### A.4 PROOF OF THEOREM 3

*Proof.* Let $y_t = x_t - \eta\nabla f(x_t)$ and $u_t \equiv y_t$. At iteration $t$, the $k$-means shortlist with envelope bounds returns a candidate $\hat{e}_t \in \mathcal{C}$ together with a *score certificate* $\varepsilon_t^{\mathrm{cert}} \geq 0$ satisfying

$$\langle u_t, e^\star \rangle - \langle u_t, \hat{e}_t \rangle \leq \|u_t\|\,\varepsilon_t^{\mathrm{cert}}, \qquad e^\star \in \arg\max_{e \in \mathcal{C}} \langle u_t, e \rangle. \tag{16}$$

(Equivalently, $e^\star$ is the exact projection of $u_t$ onto $\mathcal{C}$ since $\|e\| = 1$ for $e \in \mathcal{C}$.) By Theorem 2 (Score certificate $\Rightarrow$ Q-certificate), setting $\hat{x}_{t+1} := \hat{e}_t$ we obtain the $\delta$-prox condition (equation (4))

$$\langle u_t - \hat{x}_{t+1},\, x_t - \hat{x}_{t+1} \rangle \leq \eta\,\delta_t \quad \text{with} \quad \delta_t = \frac{\|u_t\|}{\eta}\,\varepsilon_t^{\mathrm{cert}}.$$

Thus all hypotheses of Theorem 1 (Convergence with $\delta$-prox certificates) hold with this choice of $\delta_t$. Since $f$ is $L$-smooth and $\eta \leq 1/L$, applying Theorem 1 yields, after $T$ iterations,

$$\min_{0 \leq t < T} \|\hat{x}_{t+1} - x_t\|^2 \leq \frac{2\eta}{1 - \eta L} \left( \frac{f(x_0) - f^\star}{T} + \bar{\delta}_T \right), \qquad \bar{\delta}_T = \frac{1}{T} \sum_{t=0}^{T-1} \delta_t,$$

with $\delta_t = \frac{\|u_t\|}{\eta} \varepsilon_t^{\text{cert}}$ as stated. This is exactly the claim of Theorem 3. $\qquad\square$

### A.5 PROOF OF THEOREM 4

Let $\eta \leq 1/L$ and recall from Theorem 1 (general $\mathcal{C}$ case) that for any $T \geq 1$,

$$\min_{0 \leq t < T} \|\hat{x}_{t+1} - x_t\|^2 \leq \frac{2\eta}{1 - \eta L} \left( \frac{f(x_0) - f^\star}{T} + \bar{\delta}_T \right), \qquad \bar{\delta}_T := \frac{1}{T} \sum_{t=0}^{T-1} \delta_t. \tag{17}$$

Assume we target accuracy $\varepsilon > 0$ in the step-distance and let $\bar{\Delta}$ be any uniform bound such that $\bar{\delta}_T \leq \bar{\Delta}$ for all $T$. To guarantee $\min_{0 \leq t < T} \|\hat{x}_{t+1} - x_t\|^2 \leq \varepsilon^2$, it suffices by equation 17 that

$$\frac{2\eta}{1 - \eta L} \left( \frac{f(x_0) - f^\star}{T} + \bar{\delta}_T \right) \leq \varepsilon^2 \impliedby \frac{2\eta}{1 - \eta L} \left( \frac{f(x_0) - f^\star}{T} + \bar{\Delta} \right) \leq \varepsilon^2.$$

Rearranging gives

$$\frac{f(x_0) - f^\star}{T} \leq \frac{1 - \eta L}{2\eta} \varepsilon^2 - \bar{\Delta}.$$

The right-hand side is positive precisely when

$$\varepsilon^2 > \frac{2\eta}{1 - \eta L} \bar{\Delta},$$

which is the stated condition. Under this condition we obtain the iteration bound

$$T \geq \frac{f(x_0) - f^\star}{\dfrac{1 - \eta L}{2\eta} \varepsilon^2 - \bar{\Delta}} \,.$$

$\qquad\square$

## B ETHICS STATEMENT

This work adheres to the ICLR Code of Ethics. Our study does not involve human subjects, personally identifiable information, or sensitive user data. All datasets used are publicly available and released under licenses that permit research usage. We have carefully considered potential ethical concerns, including fairness, bias, and possible misuse. The research does not involve applications that could cause harm to individuals or groups, nor does it raise issues related to privacy, security, or legal compliance. There are no conflicts of interest or external sponsorships influencing the research outcomes.

## C REPRODUCIBILITY STATEMENT

We have made significant efforts to ensure the reproducibility of our results. All model architectures, training procedures, and hyperparameters are described in detail in Sections "TOY EXAMPLE EXPERIMENT" of the paper. Full proofs of theoretical results are given in "APPENDIX: PROOFS OF LEMMAS AND THEOREMS". To further support reproducibility, we will release anonymized source code and scripts for training and evaluation as supplementary materials. Our experiments can be replicated with the information provided in the main paper, appendix, and supplementary code.