# OpenReview forum: "K-PGD: Fast Discrete Projected Gradient Descent with K-Means Acceleration on GPT"
_ICLR.cc/2026/Conference — Submitted to ICLR 2026_

### Official Review · Reviewer_nLvy · 2025-10-15

**Soundness:** 2
**Presentation:** 2
**Contribution:** 2
**Rating:** 2
**Confidence:** 3

**Summary:**

This paper studies efficient algorithm for projected gradient descent (PGD) when the constraint set is discrete. The main runtime bottleneck of prior works is the expensive projection step, where one has to search through the entire discrete set to compute the projection. To improve the efficiency, this work proposes to first perform a $k$-means clustering on the discrete set, then observe that the PGD update is to search for the point that maximizes the inner product, which can be done by first locating the top-$M$ clusters, then doing a search only insides these clusters. Theoretically, authors show that this approach only incurs a $\delta_t$ additive error to the surrogate loss, which in turn gives standard convergence bounds with additive error. Experiments are performed to show that compare to vanilla PGD, this $k$-means based PGD improves the runtime (sometimes), but surprisingly improves the success rate of adversarial attack.

**Strengths:**

* Improve the efficiency of PGD over a discrete set is an important problem, as the vanilla formulation has many room to be accelerated.

* Elementary theoretical analysis is performed to show that $k$-means based approach ensures the convergence up to an additive error term.

* Experiments are performed to show that effectiveness of the proposed method, in particular the improved success rate of adversarial attack.

**Weaknesses:**

* The underlying algorithmic question is quite simple --- given a bunch of embedding to be preprocessed, develop a data structure that supports fast retrieval of an embedding that approximates the maximum inner product of the query. This is a very standard max inner product search data structure task, and has been used in the context of Frank-Wolfe [1] and SGD [2]. Given the nature of this problem, I would strongly encourage authors to compare with algorithms that utilize max inner product search data structures (e.g., the existing implementation FAISS [3]), as this seems to be the first obvious comparison in addition to the vanilla PGD algorithm. Efficiency-wise, I would guess that algorithm using max ip data structures would be much faster than the $k$-means variant.

* The theoretical contribution is quite preliminary. Proofs are quite straightforwardly adapted from standard analysis with one small additive error twist. In particular, the additive error term $\epsilon_t^{\rm cert}$ can be large if not for all embeddings are normalized; in some applications such as text classifiers and token-level tasks, embeddings are unnormalized, which would greatly hinder the proposed framework.

* Experiments lack several important aspects. As the algorithm is clustering-based, it is imperative to understand the structure of the data, e.g., how clustered are the data? If one runs the vanilla PGD, what's the ratio that the desired projection lies in the clusters of the top-$M$? One would expect the performance of the proposed algorithm to be much better on well-clustered data, and poorly on data without such structure. Moreover, many details of the experiment are remained unexplained, for example, what is the choice of hyperparameter $M$? Are the number of iterations fixed or varied for vanilla and $k$-means PGD? Also, how can the $k$-means based algorithm is sometimes slower than the vanilla algorithm? Is it because the preprocessing takes too much time? What is the per iteration cost not accounting for preprocessing? Sometimes, data structure-based acceleration only kicks in when the size of the dataset is large enough, and the experiments of this paper are performed on fairly small datasets. It might be good for authors to try larger ones. Finally, it is surprising that using an approximate data structure would yield better successful attack rate, this is clearly not reflected and not well-explained by the theory developed in this paper (the theory merely suggests using the $k$-means doesn't hinder the convergence too much), and the improved success rate is across the board for all choices of $K$, it seems to point to the fundamental sub-optimality of the PGD algorithm.

[1] Breaking the Linear Iteration Cost Barrier for Some Well-known Conditional Gradient Methods Using MaxIP Data-structures. Zhaozhuo Xu, Zhao Song, Anshumali Shrivastava. NeurIPS'21.

[2] Fast and Accurate Stochastic Gradient Estimation. Beidi Chen, Yingchen Xu, Anshumali Shrivastava. NeurIPS'19.

[3] Billion-scale similarity search with GPUs. Jeff Johnson, Matthijs Douze,  Herve Jegou. IEEE Transactions on Big Data'21.

**Questions:**

See weaknesses.

---

### Official Review · Reviewer_3qn4 · 2025-10-23

**Soundness:** 2
**Presentation:** 2
**Contribution:** 2
**Rating:** 4
**Confidence:** 3

**Summary:**

This paper studies the design of projected gradient descent (PGD) onto a discrete set, with a focus on adversarial attacks on language models. The key idea is to accelerate the projection step by reducing it to a nearest neighbor search (NNS) problem over token embeddings, and then further speed it up using a k-means clustering–based shortlist. The paper provides a convergence guarantee for this “inexact projection” scheme and evaluates the method (IKMP-accelerated PGD) on various text classification tasks with GPT-2.

**Strengths:**

- **Timely topic**. Adversarial PGD attacks on large language models (LLMs) are an active area of research, with growing practical importance for evaluating model robustness.

- **Theoretical contribution**. The authors provide a formal certificate showing convergence under accumulated inexactness in the projection step.

- **Experiments**. The paper combines theoretical convergence analysis with empirical validation, reporting multiple evaluation metrics — including Token Error Rate (TER), Average Cosine Similarity (ACS), Success Attack Rate (SAR), and Attack Time — across several text classification datasets.

- **Clarity of presentation.** Figure 1 and the pseudo-code are clearly presented, making it relatively easy to follow the algorithmic design.

**Weaknesses:**

- **Presentation mismatch**. The abstract and contribution section are not fully aligned: the abstract frames the contribution as “optimizing PGD over a discrete set,” while later sections mix in additional aspects. The introduction also does not clearly motivate why this particular PGD attack is important compared to existing text attack strategies.

- **Novelty in the method**. The core technical idea is essentially turning discrete PGD projection into a nearest neighbor search problem over a large vocabulary. While the k-means acceleration is neat, hierarchical and clustering-based NNS methods are already well-studied in the similarity search literature. As a result, the reviewer is not convinced by the method's novelty.

- **Experimental scope**.  The tasks are limited to text classification; no evaluation is provided on generative tasks, where discrete projection may behave differently.

- **Standard theory**. The theoretical analysis is correct but relatively standard; the convergence proof is not particularly difficult or novel.

- **Practical limitations**. If the discrete set is very large, the preprocessing cost for k-means clustering can become significant, which raises concerns about scalability.
- **Missing baselines**. The paper does not compare against similarity search libraries or quantization-based projection methods, which are natural baselines for this kind of problem.

**Questions:**

Q1: When the paper mention “certificated strength”, is it referring to certified robustness guarantees? Please clarify what exactly is "strength" here.

Q2: Why not compare with existing similarity search methods (e.g., FAISS, hierarchical k-means, graph-based NNS)? There is extensive literature in this area using hierarchical clustering that seems directly relevant.

Q3: Why is the approach evaluated only on classification tasks? Would it fail or need modification for generative tasks, where the projection might affect the autoregressive token distribution?

Q4: Why is there no comparison with quantization methods, especially since projecting onto a discrete embedding/codebook set is conceptually related to quantized model inference?

---

### Official Review · Reviewer_hBeh · 2025-10-28

**Soundness:** 3
**Presentation:** 3
**Contribution:** 2
**Rating:** 4
**Confidence:** 3

**Summary:**

The paper proposes a computationally efficient projected gradient descent suitable for optimizing over a dictionary or discrete set. The idea is to first preprocess the dataset and group it into $K$ clusters using $k$-means clustering. Then, in each iteration after performing the gradient descent step, there is an inexact projection step. In this step, the top $M$ clusters are selected based on the centroids and radii of each cluster, and the projection is carried out only within these clusters, reducing computational complexity. The approach thus trades off exactness for speed, aiming to maintain good convergence behavior while significantly lowering the cost of each projection.

**Strengths:**

The paper provides a convergence guarantee for the proposed method. It shows that when the approximation error in the inexact projection step is properly controlled, the algorithm attains the same convergence rate as in the exact projection case. Moreover, it derives an explicit bound on the approximation error in terms of the centroids and radii of the clusters.

Developing a computationally efficient projected gradient descent is essential, as the main bottleneck of such algorithms typically arises in the projection step, especially when optimizing over discrete sets.

**Weaknesses:**

A key weakness of the paper lies in the absence of a mechanism for controlling the approximation error in the inexact projection step. Although the paper introduces the certificate $\epsilon_t^{\text{cert}}$ to measure this error, it does not provide an explicit procedure for selecting the keep and ignore sets in a way that guarantees $\epsilon_t^{\text{cert}}$ remains bounded or decreases over time. The method simply fixes the number of top clusters $M$, without any adaptive adjustment based on the certificate or cluster geometry. As a result, the algorithm may yield large projection errors in some iterations, preventing tight control of the bias introduced by the inexact projection.

Moreover, the convergence guarantees do not ensure that the algorithm approaches the true optimum, as the derived results only hold up to an additive error term that depends on the average approximation error. When $\epsilon_t^{\text{cert}}$ is not sufficiently small, this residual term can be significant, leading the algorithm to converge to a noticeably suboptimal point. In such cases, the resulting “flat” convergence region may be large, meaning the method could stabilize far from the optimal solution even though the theoretical bound remains valid.

**Questions:**

In the complexity analysis, you state that the per-iteration cost of the proposed method is $O(Kd + (MV/K)d)$, which seems to assume that the clusters are approximately balanced, i.e., $|V_j| \approx V/K$. Could you clarify how this complexity would change in cases where the clusters are highly unbalanced?

---

### Official Review · Reviewer_36b7 · 2025-10-31

**Soundness:** 3
**Presentation:** 1
**Contribution:** 2
**Rating:** 2
**Confidence:** 4

**Summary:**

This paper applies an approximate nearest neighbor (ANN) method to the PGD problem, providing theoretical justification through $\delta$-proximal convergence analysis and some experiments. This is a useful contribution because the projection step of PGD often dominates the gradient update. However, the proposed method overlaps almost entirely with existing ANN techniques.

Given the weaknesses, I recommend the paper is rewritten to compare (in theory and practice) the wide variety of approaches from the ANN literature when applied to the PGD problem, with the goal of finding the ANN method with the best tradeoffs for the PGD problem.

**Strengths:**

The main strength of this paper is that it recognises that PGD can benefit from ANN acceleration, in theory and practice.

**Weaknesses:**

There are two critical issues that need addressing. Firstly, the paper doesn’t mention existing ANN methods. Some of these techniques are far beyond what is being proposed here. Indeed, the use of k-means to pre cluster the input is identical to the standard Inverted File Index method that can be found in popular ANN toolkits such as FAISS. Existing methods already provide sublinear search with theoretical recall guarantees.

Secondly, the paper does not address the relationship between the number of clusters, the quality of the convergence guarantee, and runtime. This is critical to understanding how the algorithm will perform in theory and practice. If k is constant, then large clusters of size $\Omega(V)$ are possible, which will destroy the claimed per iteration time of O(Kd + (MV/K)d) since it will become O(Kd + Vd), just as slow as naive PGD.

**Questions:**

What is the relationship between the number of clusters and the convergence guarantees?

How does the proposed method compare to other ANN approaches?

---

### Meta-Review · Area_Chair_uy6J · 2026-01-07

**Summary:**

the authors did not respond to any review, even before the incidents at this year's ICLR, given the scores of the paper, I find it hard to recommend anything else but rejection

In particular, reviewers pointed several issues that deserve answers from the authors:

- Reviewer 36b7 asks about the overlap between the paper's method and an existing approximate nearest neighbor method
- Reviewer hBeh raised an important question on the absence of a mechanism for controlling the approximation error in the inexact projection step, the question remained unaddressed.
- Reviewer nLvy asked important questions on practicality and experiments' lack of clarity
- Reviwer 3qn4 points to several weaknesses in both presentation and core contribution, also remained unaddressed.

**Reviewer Concerns:**

None of the reviewers' concern was addressed by the authors during the rebuttal period.

**Reviewer Scores:**

Given that the authors did not reply to any review, I do not believe that reviewers would have changed their scores in this case, in particular, reviewer 36b7's concern on almost entire overlap with an existing method deserve a detailed answer, which the author did not address.

---

### Decision · Program_Chairs · 2026-01-26

Reject